# Determinants of Burnout among Teachers: A Systematic Review of Longitudinal Studies

**DOI:** 10.3390/ijerph19095776

**Published:** 2022-05-09

**Authors:** Dragan Mijakoski, Dumitru Cheptea, Sandy Carla Marca, Yara Shoman, Cigdem Caglayan, Merete Drevvatne Bugge, Marco Gnesi, Lode Godderis, Sibel Kiran, Damien M. McElvenny, Zakia Mediouni, Olivia Mesot, Jordan Minov, Evangelia Nena, Marina Otelea, Nurka Pranjic, Ingrid Sivesind Mehlum, Henk F. van der Molen, Irina Guseva Canu

**Affiliations:** 1Institute of Occupational Health of RNM, WHO Collaborating Center, 1000 Skopje, North Macedonia; minovj@hotmail.com; 2Faculty of Medicine, Ss. Cyril and Methodius, University in Skopje, 1000 Skopje, North Macedonia; 3Faculty of Medicine and Pharmacy, Nicolae Testemitanu State University of Medicine and Pharmacy, 2004 Chisinau, Moldova; dumitru.cheptea@usmf.md; 4Center of Primary Care and Public Health (Unisanté), University of Lausanne, 1066 Epalinges-Lausanne, Switzerland; marca.sandy@gmail.com (S.C.M.); yara.shoman@unisante.ch (Y.S.); zakia.mediouni@unisante.ch (Z.M.); mesot.olivia@gmail.com (O.M.); irina.guseva-canu@unisante.ch (I.G.C.); 5Department of Public Health, Faculty of Medicine, Kocaeli University, İzmit 41001, Turkey; cigdem.caglayan@gmail.com; 6National Institute of Occupational Health (STAMI), 0363 Oslo, Norway; mdb@stami.no (M.D.B.); ingrid.s.mehlum@stami.no (I.S.M.); 7Department of Public Health, Experimental and Forensic Medicine, University of Pavia, 27100 Pavia, Italy; marco.gnesi@unipv.it; 8Department of Primary Care and Public Health, University of Leuven, 3000 Leuven, Belgium; lode.godderis@kuleuven.be; 9Department of Occupational Health and Safety, Institute of Public Health, Hacettepe University, Ankara 06100, Turkey; sibelkiran@gmail.com; 10Research Group, Institute of Occupational Medicine, Edinburgh EH14 4AP, UK; damien.mcelvenny@iom-world.org; 11Centre for Occupational and Environmental Health, University of Manchester, Manchester M13 9PL, UK; 12Medical School, Democritus University of Thrace, 68100 Alexandroupolis, Greece; enena@med.duth.gr; 13Clinical Department 5, Carol Davila University of Medicine and Pharmacy, 020021 Bucharest, Romania; dr.marinaotelea@gmail.com; 14Department of Occupational Medicine, School of Medicine, University of Tuzla, 75000 Tuzla, Bosnia and Herzegovina; pranicnurka@hotmail.com; 15Clinic of Occupational Pathology and Toxicology, University Institute of Primary Health, 75000 Tuzla, Bosnia and Herzegovina; 16Institute of Health and Society, University of Oslo, 0373 Oslo, Norway; 17Amsterdam UMC Location University of Amsterdam, Public and Occupational Health, Netherlands Center for Occupational Diseases, Meibergdreef 9, 1100 DD Amsterdam, The Netherlands; h.f.vandermolen@amsterdamumc.nl; 18Amsterdam Public Health Research Institute, Societal Participation & Health, 1105 BP Amsterdam, The Netherlands

**Keywords:** burnout, predictors, exhaustion, teachers, occupational health, prevention

## Abstract

We aimed to review the determinants of burnout onset in teachers. The study was conducted according to the PROSPERO protocol CRD42018105901, with a focus on teachers. We performed a literature search from 1990 to 2021 in three databases: MEDLINE, PsycINFO, and Embase. We included longitudinal studies assessing burnout as a dependent variable, with a sample of at least 50 teachers. We summarized studies by the types of determinant and used the MEVORECH tool for a risk of bias assessment (RBA). The quantitative synthesis focused on emotional exhaustion. We standardized the reported regression coefficients and their standard errors and plotted them using R software to distinguish between detrimental and protective determinants. A qualitative analysis of the included studies (*n* = 33) identified 61 burnout determinants. The RBA showed that most studies had external and internal validity issues. Most studies implemented two waves (W) of data collection with 6–12 months between W1 and W2. Four types of determinants were summarized quantitatively, namely support, conflict, organizational context, and individual characteristics, based on six studies. This systematic review identified detrimental determinants of teacher exhaustion, including job satisfaction, work climate or pressure, teacher self-efficacy, neuroticism, perceived collective exhaustion, and classroom disruption. We recommend that authors consider using harmonized methods and protocols such as those developed in OMEGA-NET and other research consortia.

## 1. Introduction

Across different countries, job stress among teachers has been recognized as a common problem, receiving a significant research attention [1,2,3,4]. While the studies indicate that burnout fluctuates between and within individuals, there is a lack of evidence concerning how burnout develops over time and whether it represents a long-term or short-term condition [5]. It has been estimated, for example, that a wide range of U.S. teachers (between 5% and 20%), regardless of level, exhibit burnout [6], indicating the stressful nature of the teaching occupation [7,8]. In a cohort of 310 Swedish school teachers, it was also shown that substantial proportions of teachers showed signs of burnout, at 14% and 15%, respectively, measured at two time points 30 months apart [9]. In addition, a study conducted in Finnish teachers detected that burnout mediated the effects of high job demands on ill health [10].

Even though there are numerous studies reporting the high prevalence of stress among teachers [11,12,13], often accompanied by exhaustion and cynicism [14], there are also studies demonstrating teachers’ enthusiasm and job satisfaction [8,15,16].

### 1.1. Work Context and Exposure in the Teaching Profession

According to the International Standard Classification of Occupations (ISCO-08), teaching professionals are classified into: University and Higher Education Teachers, Vocational Education Teachers, Secondary Education Teachers, Primary School and Early Childhood Teachers, and Other Teaching Professionals [17]. The essential activity of a teacher is to enable students’ learning. To achieve teaching goals, teachers prepare lessons and exercises, develop learning materials, lead students throughout the curriculum, grade students’ work, give feedback, and collaborate with colleagues and school leaders [18]. 

The work context in which teachers provide their educational activities is very specific. The teachers in elementary schools typically work with the same group of students every day and teach students several subjects. On the contrary, high school teachers usually work with different groups of students, focusing their teaching work on one or two subjects. Apart from giving lectures, the teachers also occasionally have to meet with parents. Communicating with parents is an important part of their job, especially when students are struggling and need extra help or attention outside of the classroom. University and higher education teachers teach their subjects after the secondary education has been finalized, in addition to conducting research and preparing scholarly papers and books. Vocational education teachers teach vocational or occupational subjects in adult and further education institutions, in addition to teaching senior students [17].

A significant number of workplace hazards that teachers are exposed to have already been recognized and well-defined. These include physical (e.g., inadequate ambient temperature), biological (e.g., bacteria, viruses, and mold), and chemical exposures (e.g., laboratory chemicals), as well as ergonomic hazards (e.g., sitting or staying in one place for long periods of time).

The most common health conditions in teachers resulting from occupational exposure can be summarized into: musculoskeletal disorders due to job stress, besides ergonomic factors [19]; voice disorders resulting from vocal overload in the presence of background noise [20]; and mental health problems as a consequence of psychosocial hazards (e.g., increased job demands or limited job resources) [21]. Of note, burnout is one of the most frequently studied adverse effects of psychosocial exposures at work. Indeed, specific job demands in the teaching profession, especially increased responsibilities and tight deadlines, identify teaching as one of the most stressful occupations [7,22,23,24].

It is well known that exposure to chronic workplace stressors can result in development of burnout [25,26]. The job demands–resources (JD/JR) model of burnout assumes that the workplace context is characterized by a variety of physical, psychological, social, or organizational factors (also referred as job demands) that require prolonged physical or psychological efforts in workers. Job demands are not necessarily negative, but they may turn into workplace stressors when the invested personal efforts are high, meaning job demands may be associated with certain physiological or psychological costs [27], leading to overtaxing and emotional exhaustion. Additionally, the lack of job resources may result in withdrawal behavior (depersonalization) and disengagement [26,28,29]. Accordingly, job resources are those aspects of the job that reduce job demands (and the associated costs) and stimulate personal growth and learning and development [27]. In the context of reduced job resources (e.g., inappropriate performance feedback, low salary, job insecurity, inadequate supervisory coaching and teamwork), job demands are particularly detrimental [28,29,30].

### 1.2. Burnout in Teachers: Current State of Knowledge

Schaufeli and Taris [31,32,33] conceptualized burnout as the “combination of the inability and unwillingness to spend the necessary effort at work for proper task completion”. In this context, inability and unwillingness are two inseparable components, representing energetic and motivational dimensions, respectively [34]. 

Three dimensions (exhaustion, cynicism, and lack of professional efficacy) usually define burnout, but evidence indicates that lack of professional efficacy plays a divergent role as compared to exhaustion and cynicism [30,35]. Empirical results confirm the exceptional role of lacking efficacy compared with the other two burnout dimensions, illustrated by: a low correlation of lack of efficacy with exhaustion and cynicism; findings that burnout manifests itself via exhaustion and cynicism in psychotherapeutic clients, but is not manifested by lacking efficacy; and evidence that lack of efficacy shows a different pattern of correlations with job characteristics when compared with exhaustion and cynicism [36].

Exhaustion at both physical and psychological levels constitutes the core dimension of occupational burnout. According to the harmonized definition of occupational burnout elaborated within the framework of the EU COST Action CA16216 (The Network on the Coordination and Harmonization of European Occupational Cohorts—OMEGA-NET; http://omeganetcohorts.eu, accessed on 5 May 2022), it is characterized as a state of physical and emotional exhaustion due to prolonged exposure to work-related problems. In this EU COST Action, occupational burnout was chosen as priority health outcome for harmonization of its definition, measurement, and research protocols for future epidemiological studies. To do so, systematic reviews have been conducted [37,38,39,40], including on the predictors of occupational burnout onset, regardless of the type of occupation activity or job [41]. 

In this article, a systematic review on determinants of occupational burnout among teachers is presented. We include only longitudinal studies, since cross-sectional studies do not consider temporality.

### 1.3. Objective 

The objectives of this systematic review focused on longitudinal studies were to identify the determinants of burnout onset in teachers and to show how much further beyond qualitative analysis we can go with quantitative synthesis.

## 2. Methods

The study was conducted according to a study protocol registered in PROSPERO (CRD42018105901), with a focus on teachers. We performed a literature search for the period 1990 (January) to 2018 (August) in three databases: MEDLINE, PsycINFO, and Embase. Because there was a possibility that additional studies were published during or after finalizing this review, we checked databases for new publications up until December 2021, and no additional prospective longitudinal studies on burnout in teachers were identified.

### 2.1. Study Selection and Criteria for Inclusion and Exclusion

We included only prospective longitudinal studies where burnout was a dependent variable (outcome), with a final sample of at least 50 teachers per exposure group (i.e., studies with sufficient power), published between the years 1990 (January) and 2018 (August) in peer-reviewed journals, with no language limitation. The full search strategy can be found in Appendix A. In cases where we identified multiple publications describing a single study, we included the study only once and chose the most complete or most recent publication. We excluded studies where exposure was not assessed prior to the outcome assessment. Other reasons for exclusion were: no full text available; and studies where participants were not professionally employed (e.g., students). For one study, written in German [42], we could not assess the risk of bias (RoB) and excluded this study at this stage.

Titles and abstracts were screened independently by two reviewers (D.C. and S.C.M.) against the inclusion criteria. For studies that were not excluded on the basis of the title or abstract, full-text manuscripts were obtained and assessed by two reviewers. Any discrepancies were resolved through discussion, and if required a third reviewer (IGC) was consulted.

### 2.2. Data Extraction and Quality Assessment

We developed a MS Excel data extraction form, which was tested by all reviewers and improved until reaching consensual approval. From each included study, two reviewers (D.M. and D.C.) independently extracted the data as follows: study reference, country, objective, design, hypothesis tested and result (confirmed or not confirmed hypothesis), burnout definition used, tool used for burnout measurement, inclusion and exclusion criteria, number of occupational groups for which separate data are available, number of samples used, initial sample size, final sample size of occupational group, sex ratio (F/M), mean age (min/max), setting (urban/rural), participation rate, participation rates for other waves if several (max 4 waves), time interval between waves, number of burnout domains investigated, definition of predictive (explanatory) variables (including burnout determinants), statistical method used, type of result reported (e.g., slope or relative risk with differences across burnout measures), final model description, burnout domain name, wave number, occupational group, category of predictor and outcome (cut-off values of category boundaries, ordinal level, or continuous variable), result, result variability measure (e.g., SE, CI, *p*-value), variability value, and author’s interpretation. Particular attention was paid to the outcome definition and assessment, along with the precision of the method or tool used and the cut-off values. The same attention was paid on the determinants studied. 

### 2.3. Risk of Bias Assessment

The RoB was assessed using the Methodological Evaluation of Observational Research (MEVORECH) [43], which automatically produces a RoB report in MS Access format, taking into account the external validity (sampling of subjects, assessment of sampling bias, response rate, exclusion rate from the study, reported methods used to address sampling bias) and internal validity of the study (methods used to obtain data on dependent and independent variables, reference period, reported validation, and reliability of used methods; treatment of confounding factors; data on the loss of follow-up, appropriateness of the used statistical methods, reporting of the tested hypotheses, precision of the estimates, and sample size justification). 

### 2.4. Qualitative and Quantitative Synthesis

Studies were summarized in a narrative synthesis with two summary tables: per study and per type of independent variable (determinants). All determinants were grouped into 4 types according to the previous studies, knowledge, and experience as follows: support, conflict, individual characteristics, and organizational context. Support is defined as “an interpersonal transaction of help from a support source to the help receiver that involves emotions, material assistance, and information and that takes place in a specific family, work, or care-giving context” [44]. Conflict refers to a workplace phenomenon that is not harmful when handled objectively and in a timely manner but that leads to lost communication, affecting people and work performance when not handled at all or handled wrongly [45]. In addition to other factors, individual characteristics, such as perceived self-efficacy, coping strategies, sense of defeat, or demographic characteristics, could be related to burnout. The organizational context of burnout is described by the areas of work life, including workload, reward, fairness, and values [46].

The ***support*** category included support from colleagues, support from a supervisor, support from the community, emotional support, and social facilitators [47,48,49]. The ***conflict*** determinant group comprised four factors: conflict with colleagues, emotional strain, parent criticism, and obstacles from parents or students. All of these factors were identified in two studies [48,49]. By reading about how each of those factors were defined, we decided to remove emotional strain from our list, as it represented the same construct as emotional exhaustion, which was not taken into account as a predictor but only as an output. Conflict with colleagues and parent criticism were found in the study by Feuerhahn et al. [48], and we selected their beta coefficients and standard errors in the most complete model. In the study by Salanova et al. [49], a single model predicting emotional exhaustion was available, which we took the data from. We ended up with three data points.

The category of ***individual characteristics*** comprised seven determinants: emotional dissonance, teacher self-efficacy (in managing student behavior), emotional exhaustion at the first wave of data collection (T1), depersonalization at T1, cynicism at T1, neuroticism (defined as “individuals who score high on neuroticism are more likely than average to be moody, and to experience such feelings as anxiety, worry, fear, anger, frustration, envy, jealousy, guilt, depressed mood, and loneliness”) [50], and job satisfaction. Each of those determinants was tested in one of the four following studies, namely the studies by Feuerhahn et al. [48], Malinen et al. [51], Salanova et al. [49], and Goddard et al. [52], except for teacher self-efficacy, which was tested in two different studies.

Furthermore, given the fact that most studies have investigated emotional exhaustion as the main dimension of burnout, this was our primary focus of the study. Additionally, and since not all studies provided the standard error value but only the *p*-values, the next step was to transform the values using a two-tailed standard normal Z-table. We divided the beta regression coefficient by the z-score and obtained the standard error. We standardized each beta regression coefficient by dividing them by their standard error, and the obtained data points were plotted using R.

For the ***organizational context*** category, in the table of determinants we included 10 factors: time pressure, classroom disruption, perceived collective exhaustion, perceived collective cynicism, workload stressors, technical obstacles, effective class management, work climate (pressure), supportive school climate, and collective teacher efficacy. The factors were distributed between five articles [48,49,51,52,53].

Besides the narrative synthesis, we summarized quantitative results reported for different determinants, considering the effect estimate (a standardized slope, for example, structural equation modeling, multiple linear regression, hierarchical linear regression, or random coefficient modeling) and variability estimate (SE, CI, or at least sample size and exact *p*-value of the chosen model). The reported estimates were standardized (Appendix A) and plotted using R software to distinguish between harmful and protective determinants, following the same procedures as in the general review [41].

With respect to the outcome, in most studies the three burnout dimensions were explored separately, while in others only emotional exhaustion was studied. Therefore, our synthesis of qualitative estimates focused on emotional exhaustion, the core dimension of burnout

## 3. Results

### 3.1. Description of Selected Studies

The PRISMA statement flowchart (Figure 1) describes the literature screening, study selection, and reasons for exclusion. The extracted data from the selected articles were stored in a table that included 240 articles for which the abovementioned variables, if available, were reported. After precise study selection and exclusion of full-text articles that described studies not conducted in teachers or studies not in compliance with inclusion or exclusion criteria, a total of 33 studies (Table 1) were included [2,47,48,49,51,52,53,54,55,56,57,58,59,60,61,62,63,64,65,66,67,68,69,70,71,72,73,74,75,76,77,78,79] and reviewed. Table 1 refers to the characteristics of 33 studies on burnout among teachers that were reviewed.

Four studies reported response rates above 60%, 8 studies between 40% and 60%, and 12 studies reported response rates below 40%. Nine studies did not provide response rates. The initial sample size varied from 78 to 5575 participants and the final simple size from 56 to 2235 persons.

Most studies implemented two waves (W) of data collection; two studies [51,70] had three waves and three studies [47,52,64] had four waves. The time period between two waves was 6 months in 8 studies and 12 months in 10 studies. Some studies used shorter (5 months [56,59,60], 3–4 months [51], or even weeks [64]), longer (21–24 months [48,55,61,66]), or similar (7–8 months [49,69,75,78,79] and 10 months [72]) periods. Parker et al. [71] did not report the reference period.

Of the 33 included studies, 13 studies used a single composite measure of burnout as a dependent variable and the other 20 studies focused on one (seven studies using emotional exhaustion (EE) as a dependent variable), two (two studies), or three (ten studies) burnout dimensions. One study [77], besides EE and personal accomplishment (PA), additionally used two dimensions of depersonalization, one regarding the students and the other one regarding their colleagues.

### 3.2. Findings of the Risk of Bias Assessment

The RoB assessment showed that most studies assessed the major confounding factors (e.g., gender, age, marital status, work experience, level of education, income), but had external and internal validity issues due to limitations in sampling, inadequate reporting of response rates and exclusion rates, or use of self-reported instruments with uncertain validity and reliability.

In more detail, the analysis of the **external validity** of the included studies (*N* = 33) within the RoB assessment demonstrated that each analyzed study was designed to detect determinants of burnout in teachers as an occupational group. Hence, all involved studies were conducted in a non-general population [2,47,48,49,51,52,53,54,55,56,57,58,59,60,61,62,63,64,65,66,67,68,69,70,71,72,73,74,75,76,77,78,79]. Similarly, self-selection of participants was demonstrated in each of the analyzed studies during a RoB assessment. Concerning the response rates across the whole sample, these were frequently either not reported [48,49,55,63,64,67,76,77,79] or were lower than 40% [47,54,56,57,58,61,62,65,66,68,72,73].

Additionally, a very frequent finding was that the exclusion (drop-out) rate from the analysis was either not reported [47,49,51,53,54,55,56,57,58,59,60,61,62,63,66,67,69,71,73,74,78,79] or was higher than 10% (e.g., failed to return the subsequent surveys, part-time work, coincidence with teachers’ strike, employment contracts not remaining the same across all measurements, missing data) [2,52,65,68,70,72,75]. Additionally, we detected that in some studies the number of screened persons [49,63,67,79] or the number of eligible individuals [49,67,79] was not reported.

The RoB assessment showed that sampling bias was not assessed [2,49,51,66,67,74], or it was not addressed in the analysis [2,47,49,54,66,67,74]. Laugaa et al. [67] did not report the sampling method that was used. Others did not justify the sample size completely [2,48,52,63,64,73].

The analysis of the **internal validity** demonstrated that several studies [2,49,51,53,56,59,60,63,64,67,68,69,72,73,75,78,79] used a reference period that was different from the one year follow-up [41,80]. In the study by Parker et al. [71], the reference period was not reported. In some studies, major confounding factors were not assessed or were only partially assessed [2,47,51,56,66,67,72,73,74,75,77,78].

Regarding the validity and reliability of measures used for assessing the independent variables, most were self-reported measures. Often authors did not report on the validity or reliability of these measures [49,53,54,55,79], while others used subscales with low reliability coefficients [51,52,75]. Rarely did we find that the intensity or dose of the exposure (independent) variable was not assessed in the study [54,55]. The authors in these studies only used binary variables, such as “presence/absence” or “yes/no”.

### 3.3. Qualitative Synthesis of Burnout Determinants in Teachers

The qualitative analysis identified 61 burnout determinants studied among teachers (Table 2). Table 2 refers to the list of burnout determinants that were detected within the qualitative analysis.

***Support*** determinants were analyzed less frequently than the other determinants (e.g., lack of social integration and work-to-family or family-to-work facilitation by Burke et al. [54] and Innstrand et al. [66], respectively). Social support from colleagues, social support from the broader community, and lack of social support or lack of work–family enrichment were examined in two or three studies.

The qualitative analysis revealed specific determinants related to ***conflict*** relationships. Hence, stress due to societal demands [59] or stress due to relationships with colleagues or the organization [76] was detected as a significant determinant in a single study, while stress due to relationship with students was detected in four studies [49,59,76,79] and stress due to parental criticism was detected in three studies [48,49,79].

Certain ***individual characteristics*** were studied in a single study (e.g., negative affectivity in the study by Houkes et al. [65], interpersonal rejection sensitivity in the study by Bianchi et al. [55], self-doubt in the study by Burke et al. [54]), whereas others were studied in several studies (e.g., perceived self-efficacy [2,48,51,56,68,74,78] or avoidance coping [67,68,71,73]).

Finally, within the category of ***organizational context*** determinants, a lack of stimulation and narrow client contacts (e.g., “I spend most of my time in my job in direct contact with other people”) were detected as significant determinants of burnout only in the study by Burke et al. [57]. On the contrary, ambiguity and conflict were detected as significant determinants in three studies [57,69,78], while time pressure, work pressure, and workload were detected in six studies [48,52,65,68,69,78].

Most of the determinants showed a detrimental effect. Work setting characteristics [58], sources of stress [58,59,68], heterogeneous classes [75], loss of status [59], downward identification [60], lack of social integration [54], and disruptive students or classroom interruptions [48,54], as well as perceived inequity in relationships with students, colleagues, and the organization [76] are some of the examples of determinants with detrimental effects. On the other hand, we also identified determinants with protective effects, such as innovation [52], school climate [51,78], perceived self-efficacy [2,48,51,56,68,74,78], or social support from colleagues [47,61] or from the broader community [47,48,68].

### 3.4. Quantitative Synthesis of Burnout Determinants in Teachers

The quantitative synthesis added a certain value to the qualitative findings. We finally obtained a list of 6 articles [47,48,49,51,52,53] that satisfied the criteria for the quantitative synthesis (Table 3). Table 3 refers to the list of burnout determinants that were analyzed via quantitative synthesis.

The three studies [47,48,49] on the ***support*** group (determinants analyzed: support from colleagues, supervisor, and community; emotional support; social facilitators) are represented individually in Figure 2. Each point represents an association between the input (in this case, support) and output (emotional exhaustion). When the coefficient is positive, a positive correlation is shown, which indicates that support increases emotional exhaustion, causing a detrimental relationship. Conversely, for a negative coefficient, a negative correlation is shown, which means that support decreases emotional exhaustion and we have a protective relationship.

In Figure 2, we first observed that there were statistically significant points only in the study by Beausaert et al. [47], and none in the two other studies. Therefore, emotional support (Feuerhahn et al. [48]) and social facilitators (Salanova et al. [49]) did not have an impact on emotional exhaustion. Concerning the study by Beausaert et al. [47], support from a supervisor was also not significant. Two support types were partially significant: support from colleagues and support from the community. The partially significant effects of both types of support meant that they were not constant over time. For colleagues’ support in both schools and community support in secondary schools, the effects were significant only in the second wave but not in the two other waves. Community support in primary school was the only one with consistent results. Finally, we observed that the effect of the community support was opposite to the colleagues’ support. Increasing the level of community support actually increases emotional exhaustion while increasing colleagues’ support diminishes emotional exhaustion.

Figure 3 presents the results from two studies [48,49] reporting the estimates for the ***conflict*** group (conflict with colleagues, emotional strain, parental criticism, and obstacles from parents or students). We observed that even though parental criticism was quite far from zero, none of the three factors tested was statistically significant.

The results for ***individual characteristics*** (emotional dissonance; teacher self-efficacy; exhaustion, depersonalization, or cynicism at T1; neuroticism; job satisfaction in studies by Feuerhahn et al. [48], Malinen et al. [51], Salanova et al. [49], and Goddard et al. [52]) are shown in Figure 4. We observed that all points except emotional dissonance had statistically significant detrimental effects on the onset of emotional exhaustion.

Figure 5 displays the results regarding ***organizational context*** determinants (time pressure, classroom disruption, perceived collective exhaustion, workload stressors, technical obstacles, effective class management, and work climate or pressure) in the studies by Feuerhahn et al. [48], Gonzalez-Morales et al. [53], Salanova et al. [49], and Goddard et al. [52]. Classroom disruption, perceived collective exhaustion, and work climate (pressure) exhibited statistically significant detrimental effects on emotional exhaustion.

## 4. Discussion

This systematic review of 33 longitudinal studies included 78 to 5575 teachers (in the first wave) and 56 to 2235 teachers (in the final sample). The studies were conducted in 11 countries. Several detrimental determinants of exhaustion were identified and classified according to their relative importance (i.e., effect size): job satisfaction as the most predictive determinant; work climate (pressure); teacher self-efficacy; neuroticism; perceived collective exhaustion; and classroom disruption as the least predictive determinant.

### 4.1. Interpretation of Findings

When interpreting the findings, one should take into consideration the risk of bias in the included studies. In this review, the risk of bias assessment resulted in a variety of methodological issues. All analyzed studies showed external or internal validity issues, mainly due to limitations in sampling, inadequate reporting of response rates and exclusion rates, or use of self-reported instruments with uncertain validity and reliability. Therefore, the sources of this bias should be carefully considered in future studies on burnout in teachers.

This systematic review showed that some burnout determinants were studied more frequently than the others (e.g., lack of stimulation and narrow client contacts in only one study; ambiguity or conflict in three studies; time pressure, work pressure, workload, or perceived self-efficacy in six or more studies). Role ambiguity [81,82,83,84] and workload or time pressure [85,86,87,88] were detected as significant burnout determinants in different settings. Perceived self-efficacy is frequently studied as a burnout determinant, mainly in teachers, but also in other occupations [89,90,91]. Surprisingly, support determinants were not so frequently analyzed (e.g., lack of social integration, work-to-family or family-to-work facilitation, social support from colleagues, social support from the broader community, lack of social support, lack of work–family enrichment), although they were frequently detected as determinants in burnout research [92,93,94,95]. Several determinants related to conflict relationships (e.g., stress due to societal demands or stress due to relationships with colleagues) were detected as significant determinants in one study, while stress due to relationships with students or parental criticism was detected in several of the analyzed studies. The interpersonal relationships and their associations with burnout were also studied in health care workers [96,97,98].

The identified determinants mostly showed detrimental effects (e.g., work setting characteristics as a composite score of inadequate orientation, workload, lack of stimulation, scope of client contacts, unclear institutional goals, lack of autonomy, poor leadership, and social isolation; sources of stress; heterogeneous classes; downward identification; lack of social integration; disruptive students or classroom interruptions; or perceived inequity in relationships with students, colleagues, and the organization). In line with the job demands–resources model of burnout [28,99], a lack of stimulation, narrow client contacts, downward identification, and ambiguity conflict are typical job demands, while a supportive school climate, autonomy, effective class management, and social support from colleagues and the broader community are classified as job resources.

The quantitative synthesis refined the importance of burnout determinants in teachers and was conducted on six selected studies. According to our observations, most of the data did not sustain the assumption that **support** is beneficial in reducing emotional exhaustion, the core dimension of burnout, although colleagues’ support seemed to be in line with the assumptions but without strong evidence. In fact, the strongest evidence contradicted the assumptions, since the community support increased the emotional exhaustion. The authors [47] noted that this is a contradictory and unexpected finding that could be explained by ‘the downside of empathy’ (i.e., principals and teachers who feel supported by the community, are more connected to the community, and also more vulnerable to the community stress). This effect was stronger in principals and teachers working in primary schools because they were part of a smaller community than those working in secondary schools, and might be more connected to the community and more affected by its stressors.

Nevertheless, we need to explain our findings. All notable effects of support were present in the study by Beausaert et al. [47], who took into account several time intervals over one year and focused on burnout over a longer period (4 years in total vs. 2 years for Feuerhahn et al. [48] and 6 months for Salanova et al. [49]). Focusing on only the first wave of this study (as was the case in the other studies), we would only have a single measurement point during follow-up. Therefore, it is necessary to question how the development goes after burnout is first detected, namely whether it improves, deteriorates, or remains stable. This will probably depend on whether any actions are taken.

Lastly, putting these results in perspective, we can consider the few statistically significant effects (positive relationship with community support and negative relationship with colleagues’ support) as suggestive rather than solid evidence. In the study by Kim et al. [100], fourteen empirical studies on burnout in students were reviewed and support was found to be negatively associated with all three burnout dimensions. Similarly, a systematic review of the work environment and burnout, not confined to any specific occupational group, showed moderately strong evidence of a relationship between low workplace support and increased emotional exhaustion [101].

Within the **conflict** group, even though parental criticism was quite far from zero (beta = 0.24, SE = 0.16, beta/SE = 0.24/0.16 = 1.5), none of the factors tested were statistically significant. As there was no significant factor in this group, we interpreted that parental criticism, conflict with colleagues, and obstacles from parents or students did not influence in any way the occurrence of emotional exhaustion. However, qualitative syntheses in the actual systematic review as well as in other studies [102,103] showed the positive relationships of obstacles with burnout. It has to be taken into consideration that the non-significant findings regarding conflict determinants in this review could be also due to the bias in these studies.

We observed that all analyzed **individual characteristics**, except emotional dissonance, were statistically significant. They all showed a positive relationship, i.e., increasing the determinant’s value increases the values of emotional exhaustion. However, the results were not exactly as expected. For the determinant “neuroticism”, it seemed logical that this individual characteristic favored the development of emotional exhaustion, and similar findings were also shown in other studies [104,105,106]. The syntheses highlighted the “emotional exhaustion at T1” as a determinant of emotional exhaustion with subsequent waves of data collection. Therefore, future studies must take into account the levels of burnout dimensions at T1 as confounding factors that are necessary to control for. We also observed positive relationships between teacher self-efficacy and job satisfaction with emotional exhaustion. These individual characteristics showed an inverse association of what we expected. Indeed, the literature has shown negative relationships between self-efficacy and job satisfaction with burnout [74,107,108]. A possible explanation for this result could be that the measure of one of those factors hides another factor. For example, a high level of job satisfaction may be reached only by working hard, implying one or more “hidden” burnout determinants (e.g., high workload). Teacher self-efficacy may also depend on how much discipline the teacher has. If the teacher does not encounter any difficulty with students, then there would be no reason for teacher self-efficacy to be especially high, as there was no “testing” occasion. Additionally, extensive indiscipline management may lead the teacher to think that they are a bad teacher and may reduce teacher self-efficacy.

The results for teacher self-efficacy were significant and consistent, showing positive relationships. This strengthened the evidence that teacher self-efficacy contributes to the emergence of emotional exhaustion. However, we cannot directly compare the two studies. Firstly, this was because teacher self-efficacy was not measured with the same tool. Feuerhahn et al. [48] used a scale developed by Schmitz and Schwarzer in 2000, while Malinen et al. [51] used the Finnish version of the Teacher Self-Efficacy for Inclusive Practices (TEIP) scale. Secondly, emotional exhaustion as an output also was not measured in the same way in both studies (Educator MBI vs. Finnish version of the Bergen Burnout Indicator 15 scale). However, the results of the two studies might still be comparable, despite using different measurement tools.

The analysis of the determinants belonging to the category of **organizational context** highlighted three significant points (perceived collective exhaustion, work climate, and classroom disruption). Perceived collective exhaustion and work climate (measured by “work pressure” or the degree to which the pressure of work and time urgency dominate the job environment) both had detrimental effects on emotional exhaustion. This seemed reasonable, as they probably were connected. Perceiving exhaustion in colleagues is part of a bad work climate. Moreover, classroom disruption could also additionally worsen the work climate. Actually, collective exhaustion and classroom disruption could be seen as “subfactors” of work climate.

Another element that caught our attention was the non-significance of the determinant “time pressure”, contrary to our initial beliefs based on the literature that it would have a detrimental effect on emotional exhaustion by increasing stress [109,110,111,112]. A possible explanation for this result could be the lower level of time pressure within the teaching sector. As the teaching occupation’s environment is not very dynamic (i.e., the activities are planned in advance), teachers are probably less exposed to time pressure than health care workers for example.

### 4.2. Methodological Considerations

This systematic review demonstrated extensive variability in the field of research of burnout determinants in teachers, with different time periods between the waves in different studies. This variability was illustrated by several points.

Firstly, the qualitative synthesis stage detected a wide range of determinants of burnout among teachers. This systematic review identified 61 factors that were highly correlated with burnout in teachers, which were studied in longitudinal settings. A wide spectrum of burnout determinants has been previously studied in teachers, not only in longitudinal studies, which we included in our systematic review, but also in cross-sectional studies [113,114]. However, quantitative synthesis revealed only six studies that had highlighted certain factors demonstrating clear significant protective or detrimental effects.

Secondly, this review highlighted the heterogeneity in the criteria used to define and measure burnout in the literature. The authors referred to a variety of statements stemming from the original definitions of burnout (Freudenberger from 1974 [115]; Cherniss from 1980 [116]; Maslach and Jackson from 1981 [14]; Shirom from 1989 [117]; Schaufeli and Enzmann from 1998 [118]; Demerouti, Bakker, Nachreiner, and Schaufeli from 2001 [28]; Kristensen, Borritz, Villadsen, and Christensen from 2005 [119]). The definition used by Maslach and Jackson from 1981 [14] was the most frequently referenced, while six studies [48,64,65,70,72,73] did not report a definition of burnout at all.

Additionally, we found that the different studies analyzed in this review used different instruments for burnout measurements. The authors mostly used different versions of the Maslach Burnout Inventory (MBI) [14], while several studies based their analyses on the Oldenburg Burnout Inventory (OLBI) [120], Copenhagen Psychosocial Questionnaire [121], Bergen Burnout Indicator [122], or Shirom–Melamed Burnout Measure [123]. Furthermore, we detected differences between studies in the measures that were used as dependent variables (e.g., composite measures of burnout; one, two, or three burnout dimensions; or two different dimensions of depersonalization).

This review identified a lack of consensus on the use of the burnout construct in measuring exposure and responses to occupational stress, with similar findings to those found in a study on physicians [124]. Different burnout definitions and different measurement tools used in research resulted in very low comparability of findings and results obtained through longitudinal studies on teacher burnout. Similar methodological heterogeneity among the studies in terms of shifting definitions of burnout and questions around the measurement tools of the burnout construct was also detected in healthcare workers [124,125,126,127,128,129] and other occupational groups [101,130,131].

Another issue that was revealed in this systematic review was the methodological inconsistency in the longitudinal studies, namely the different numbers of waves of data collection and the time periods between the waves. Similar differences were found through literature searches in other occupational groups as well [66,102,132,133,134,135,136,137,138,139,140]. It is obvious that there is a need to harmonize the coordination and development of longitudinal studies in burnout research [141]. Taking into account the latency of occupational burnout, as we have previously recommended [41], the longitudinal studies with multiple waves [142] should involve at least 12 months follow-up of exposed workers.

Within the 33 included studies, 12 (36.4%) did not control at all or only partially controlled for confounding factors. Additionally, the reference period was shorter than the one year follow-up in 17 (51.5%) studies.

It is noteworthy that the literature search did not include the gray literature due to the lack of consensus on a standardized method for searching in the gray literature, the lack of full-text studies, and the non-publishing of the gray literature in peer-reviewed journals as a quality indicator [41].

Taking into account the large number of references screened and reviewed and the multiple methodological approaches implemented in this review, there was a possibility that additional studies were published during or after finalizing this review. We checked databases for new publications up until December 2021 and no additional prospective longitudinal studies on burnout in teachers were identified.

The strengths of this systematic review study are as follows. Only longitudinal studies were included, with different durations of follow-up, since cross-sectional studies do not consider temporality [143,144]. Additionally, a focus was placed on emotional exhaustion as the main component of occupational burnout. This is a consensually accepted dimension of occupational burnout that is measured by almost all available tools, including the most valid ones [41]. As in our previously published systematic review [41], we recommend that future research considers a longitudinal design with multiple waves [142], with at least one year follow-up of exposed workers. Finally, we performed a comprehensive risk of bias assessment according to the most validated and appropriate tools (MEVORECH).

Beyond the given recommendations for future research in the field of burnout in teachers, the practical implications of this review can be seen through the identified detrimental determinants of teacher exhaustion. These factors should be targeted as a priority within the development of prevention programs on burnout in teachers. Tackling the detected determinants of teacher exhaustion could reduce the prevalence of burnout among teachers.

## 5. Conclusions

This systematic review, particularly the quantitative synthesis phase, identified several detrimental determinants (job satisfaction, work climate (pressure), teacher self-efficacy, neuroticism, perceived collective exhaustion, classroom disruption) of teacher exhaustion. These findings are especially important due to the longitudinal design of the included studies published for this occupational group over a period of 30 years. The results of the protective determinants were inconsistent between studies, varying from wave to wave, while due to the high variability in measures between studies we were not able to clearly state which determinants truly influence emotional exhaustion.

In general, the difficulty in conducting this kind of systematic review has always been the lack of harmonization in the outcome measures and study protocols in general. We recommend that authors in the future research consider using standardized methods and harmonized protocols, such as those assessed and developed in OMEGA-NET [37,38,39,145] and other research consortia.

## Figures and Tables

**Figure 1 ijerph-19-05776-f001:**
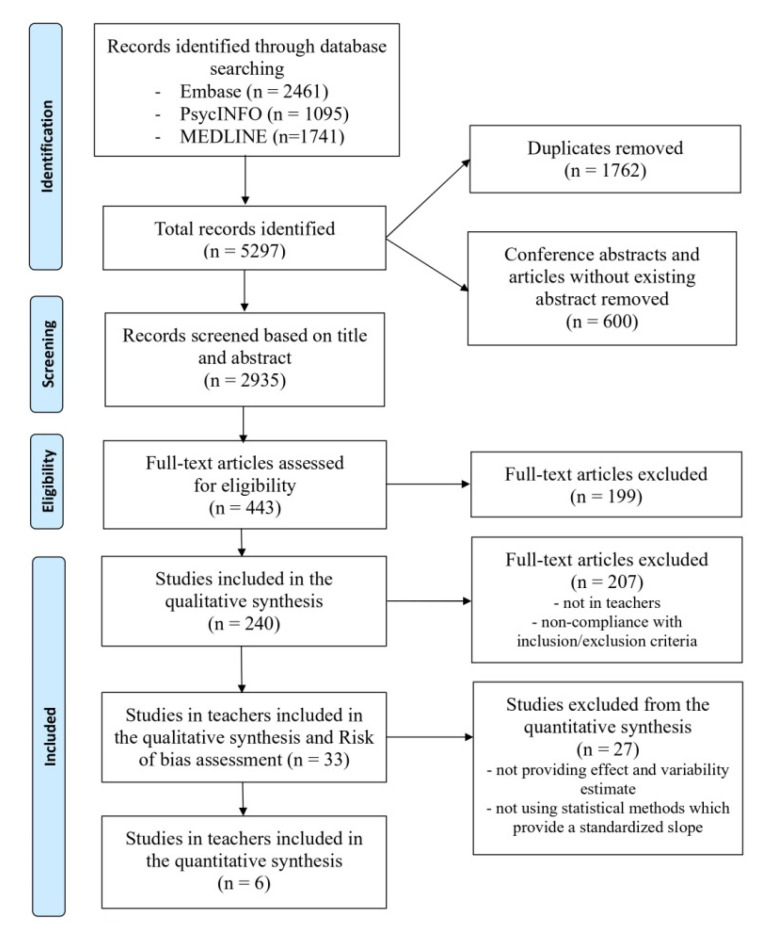
Flow chart of study identification and selection process.

**Figure 2 ijerph-19-05776-f002:**
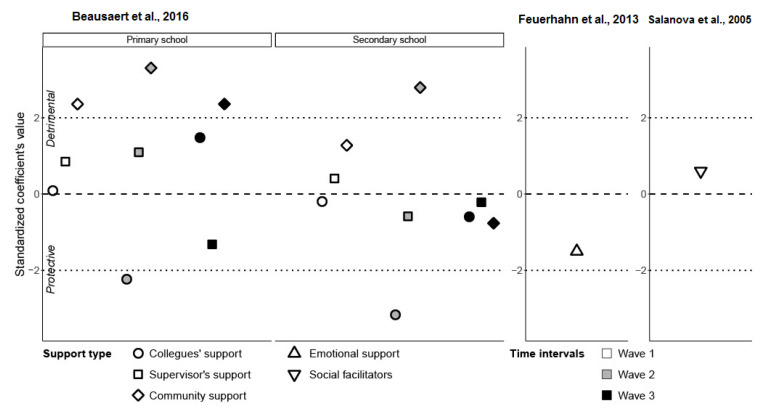
Three studies representing support category of determinants (Beausaert et al., 2016; Feuerhahn et al., 2013; Salanova et al. 2005) [47,48,49].

**Figure 3 ijerph-19-05776-f003:**
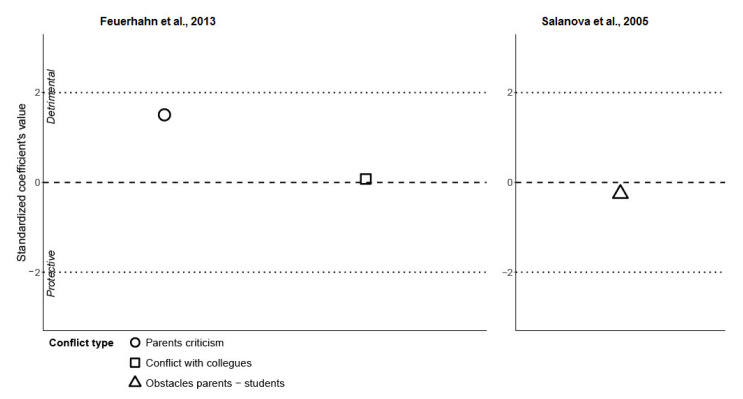
Studies showing conflict category of burnout determinants (Feuerhahn et al., 2013; Salanova et al., 2005) [48,49].

**Figure 4 ijerph-19-05776-f004:**
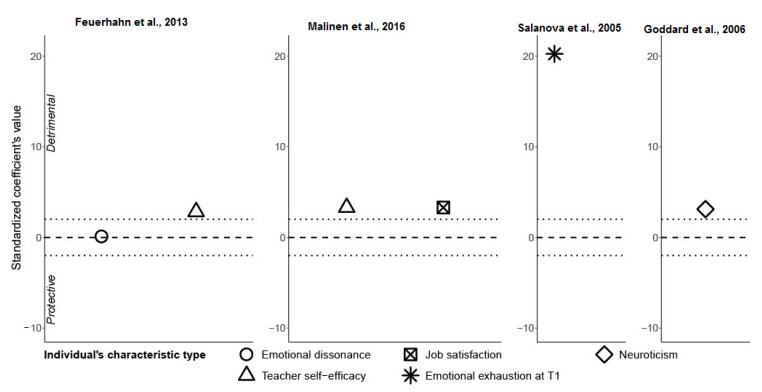
Four studies representing individual characteristics as burnout determinants (Feuerhahn et al., 2013; Malinen et al., 2016; Salanova et al., 2005; Goddard et al., 2006) [48,49,51,52].

**Figure 5 ijerph-19-05776-f005:**
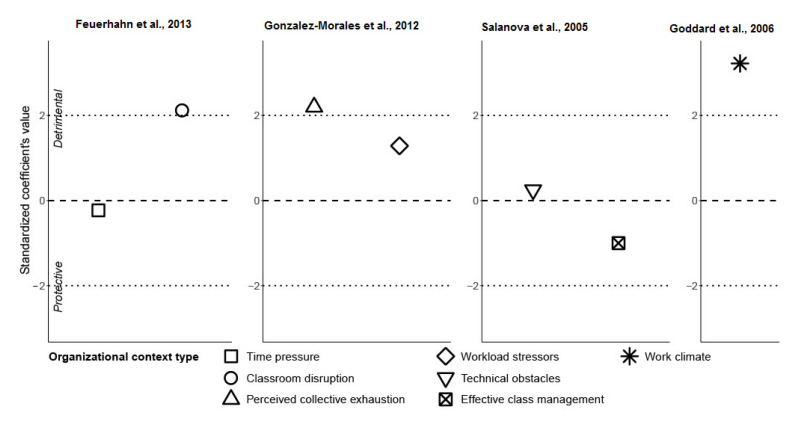
Studies showing burnout determinants belonging to the organizational context category (Feuerhahn et al., 2013; Gonzalez-Morales et al., 2012; Salanova et al., 2005; Goddard et al., 2006) [48,49,52,53].

**Table 1 ijerph-19-05776-t001:** Characteristics of studies on burnout among teachers (*n* = 33).

Study (1st Author, Journal, Year of Publication, Country) and Outcome	Follow-Up	Study Sample (*N*)	Main Significant Findings and Effects (Detrimental D or Protective P)	Risk of Bias
Beausaert et al. [47], Educational Research, 2016, Australia * Burnout	Four waves: from April to July in 2011 (T1) and from early July to late September in 2012–2014 (T2, T3, T4)	T1: 3572 T2: 20–25% T3: 20–25% T4: 20–25%	**Support** -Small negative effect of **social support from colleagues** at T2 on burnout at T3 for primary and secondary school principals–P -**Social support from colleagues** at T3 had a significant negative relationship with burnout (via stress) at T4 among the secondary school principals–P -**Support from the broader community** via the downside of empathy had a positive relationship with burnout at all measurement times (whether in primary or in both primary and secondary schools)–D -**Social support from the broader community** via stress showed significant negative indirect relationships with burnout at T1 (primary and secondary school principals) and T2 (primary school principals)–P **Individual characteristics** -Strong positive relationship between **stress** and burnout for both primary and secondary schools principals across all points of measurement–D	**External validity** *Major flaws*: -Response rate in total sample <40% *Minor flaws*: -Non-general population -Self-selection of participants -Not addressed sampling bias in analysis *Poor reporting*: -Not reported exclusion rate **Internal validity** *Major flaws*: -Major confounding factors not assessed
Bianchi et al. [55], Personality and Individual Differences, 2015, France EE and DP combined in a burnout index	Two waves: from April–June and November–December 2012 to April 2014 (mean duration of the follow-up–21 months)	T1: 5575 T2: 627	**Individual characteristics** -**Burnout symptoms** at T1 were the best predictor of cases of burnout at T2–D -Participants with **interpersonal rejection sensitivity** presented a 119% increase in the risk of being burned out–D	**External validity** *Minor flaws*: -Non-general population -Self-selection of participants *Poor reporting*: -Response rate in total sample not reported -Exclusion rate not reported **Internal validity** *Minor flaws*: -Intensity/dose of the exposure (independent) variable not assessed in the study (only presence/ absence) *Poor reporting*: -Reliability of independent variable not reported
Browers et al. [56], Teaching and Teacher Education, 2000, Netherlands EE, DP, PA	Two waves: from October 1997 to March 1998 (5 months)	T1: 558 T2: 243	**Individual characteristics** -The relationship between depersonalization and **perceived self-efficacy** showed an effect of the former on the latter, while the time frame was longitudinal–P -The relationship between personal accomplishment and **perceived self-efficacy** showed an effect of the former on the latter, while the time frame was synchronous–P	**External validity** *Major flaws*: -Response rate in total sample <40% *Minor flaws*: -Non-general population -Self-selection of participants *Poor reporting*: -Not reported exclusion rate **Internal validity** *Major flaws*: -Major confounding factors not assessed *Minor flaws*: -Reference period different from recommended
Burke et al. [57], Human Relations, 1995, NR (probably Canada) EE, DP, Lack of PA	Two waves: One year between the waves	T1: 833 T2: 362	**Organizational context** -**Lack of stimulation** showed significant and independent correlations with burnout in all cases–D -**Narrow client contacts** showed significant and independent correlation with the dependent variables in almost all cases (DP, LPA, total MBI)–D -**Conflict and ambiguity**–only with EE–D -**Type of school** was significantly correlated with depersonalization–D **Individual characteristics** -**Unmet expectations**–only with DP–D -**Individual demographic characteristics**–contributed significant levels of explained variance on EE, DP, and total MBI score–D	**External validity** *Major flaws*: -Response rate in total sample <40% *Minor flaws*: -Non-general population -Self-selection of participants *Poor reporting*: -Not reported exclusion rate
Burke et al. [58], Social Science & Medicine, 1995, NR (probably Canada) Negative attitude change (burnout) by Cherniss (composite measure) EE, DP, Lack of PA into a composite measure	Two waves: One year between the waves	T1: 833 T2: 362	**Organizational context** -**Sources of stress** and psychological burnout (positive)–D -**Lack of social support** and psychological burnout through sources of stress–D -**Work setting characteristics** and psychological burnout (positive)–D **Support** -**Lack of social support** and burnout (positive relationship)–D	**External validity** *Major flaws*: -Response rate in total sample <40% *Minor flaws*: -Non-general population -Self-selection of participants *Poor reporting*: -Not reported exclusion rate
Burke et al. [54], Anxiety Stress and Coping, 1996, NR (probably Canada) EE, DP, Lack of PA into a composite measure	Two waves: One year between the waves	T1: 833 T2: 362 *N* = 250 with complete data in final the analysis	**Organizational context** -**Red tape work** is the predictor of burnout for total sample (teachers and administrators within a single board of education), men, administrators, and teachers. The best predictor for the total sample, men, and administrators–D **Individual characteristics** -**Self-doubt** is the predictor of burnout for men–D **Support** -**Lack of social integration** is the predictor of burnout for teachers–D **Conflict** -**Disruptive students** is the predictor of burnout for total sample, women, and teachers. The best for women and teachers–D	**External validity** *Major flaws*: -Response rate in total sample <40% *Minor flaws*: -Non-general population -Self-selection of participants -Not assessed sampling bias -Sampling bias not addressed in analysis *Poor reporting*: -Not reported exclusion rate **Internal validity** *Minor flaws*: -Intensity/dose of social integration not assessed in the study (only yes/no) *Poor reporting*: -Validity and reliability of independent variables not reported
Buunk et al. [59], European Journal of Personality, 2007, Spain EE, cynicism and personal efficacy into a single measure	Two waves: twice during an academic year (first term and the third term of the academic year–5–6 months interval)	T1: 659 T2: 558	**Organizational context** -**Stress intrinsic to the job** predicted burnout at the kindergarten level–D **Individual characteristics** -In the total sample, men, and at the secondary level (low and high), a **sense of defeat** was the major predictor of a change in burnout–D -**Loss of status**–the most important predictor of burnout in women and at the kindergarten level–D **Conflict** -**Stress due to societal demands** was positively associated with burnout (total sample)–D -**Stress due to relationship with students** was positively associated with burnout (total sample, men, and high secondary school)–D -**Stress due to societal demands** predicted burnout at the kindergarten level	**External validity** *Minor flaws*: -Non-general population -Self-selection of participants *Poor reporting*: -Not reported exclusion rate **Internal validity** *Minor flaws*: -Reference period different from recommended
Carmona et al. [60], Journal of Occupational and Organizational Psychology, 2006, Spain Burnout	Two waves: twice during an academic year (first term and the third term of the academic year–5–6 month interval)	T1: 659 T2: 558	**Individual characteristics** -**Downward identification** had an independent positive relation with burnout–D -There was a significant negative effect of the use of a **direct coping style** on a change in burnout–P	**External validity** *Minor flaws*: -Non-general population -Self-selection of participants *Poor reporting*: -Not reported exclusion rate **Internal validity** *Minor flaws*: -Reference period different from recommended
Fernet et al. [61], Journal of Organizational Behavior, 2010, Canada EE, DP, PA	Two waves: 0 and 24 months	T1: 380 T2: 276 (153 new)	**Individual characteristics** -**Self-determined work motivation** was positively associated with personal accomplishment–P **Support** -High **quality of relationships with colleagues** was negatively associated with EE, DP, RPA–P **IC x Support** -**Having high-quality relationships** **with coworkers** was negatively associated with EE over time, but only for employees with **low self-determined motivation**–P -**High-quality relationships** were beneficial only for employees with **low self-determined motivation** (concerning DP)–P -**High-quality** **relationships** were more positively associated with PA over time for employees with **low self-determined motivation**–P	**External validity** *Major flaws*: -Response rate in total sample <40% *Minor flaws*: -Non-general population -Self-selection of participants *Poor reporting*: -Not reported exclusion rate
Fernet et al. [62], Work and Stress, 2014, Canada EE, cynicism, professional efficacy	Two waves: 12 month period (October to October)	T1: 1019 T2: 689	**Individual characteristics** -**Harmonious passion** had a cross-lagged effect on professional efficacy–P -**Obsessive passion** had a cross-lagged effect on emotional exhaustion–D	**External validity** *Major flaws*: -Response rate in total sample <40% *Minor flaws*: -Non-general population -Self-selection of participants *Poor reporting*: -Not reported exclusion rate
Feuerhahn et al. [63], Stress and Health, 2013, Germany EE	Two waves: 6 month period	T1: 100 T2: 87	None of the outcome variables at T1 predicted lagged EE at T2	**External validity** *Minor flaws*: -Non-general population -Self-selection of participants -Incomplete justifications of the sample size *Poor reporting*: -Response rate in total sample not reported -Number of screened not reported -Not reported exclusion rate **Internal validity** *Minor flaws*: -Reference period different from recommended
Feuerhahn et al. [48], Applied Psychology: Health and Well-Being, 2013, Germany * EE	Two waves: 21 month period	T1: 177 T2: 56	**Organizational context** -Significant main (single) effect of cognitive job demand **time pressure** on T2 emotional exhaustion–D -Significant main (single) effect of cognitive job demand **classroom interruptions** on T2 emotional exhaustion–D **Individual characteristics** -Emotional job demand **emotional dissonance** with **emotional support**–significant interaction (buffering effect)–P -Emotional job demand **emotional dissonance** with **self-efficacy**–significant interaction (buffering effect)–P **Conflict** -Emotional job demand **parents’ criticism**–significant main (single) effect on T2 emotional exhaustion–D -Emotional job demand **conflicts with colleagues** with **emotional support**–significant interaction (buffering effect)–P	**External validity** *Minor flaws*: -Non-general population -Self-selection of participants -Incomplete justifications of the sample size *Poor reporting*: -Response rate in total sample not reported
Flaxman et al. [64], Journal of Applied Psychology, 2012, United Kingdom EE	Four waves: To coincide with the 2008 Easter holiday Prerespite (T1)–one or two working weeks prior to the Easter weekend T2–during the Easter respite T3–either the first or the second full week back at work T4–fourth or the fifth full working week after the Easter weekend	T1: 111 T2: not reported T3: 100 T4: 77	**Individual characteristics** -Academics exhibiting a **self-critical form of perfectionism** were found to report significantly higher exhaustion (T3, not T4)–D -Indirect effects of **perfectionism** on respite to post respite change in exhaustion, anxiety, and fatigue via worry and rumination during the respite–D -Academics who reported greater work-related **worry and rumination** during the respite showed elevated emotional exhaustion (T3 not T4)–D	**External validity** *Minor flaws*: -Non-general population -Self-selection of participants -Incomplete justifications of the sample size *Poor reporting*: -Response rate in total sample not reported **Internal validity** *Minor flaws*: -Reference period different from recommended
Goddard et al. [52], British Educational Research Journal, 2006, Australia* EE, DP, PA	Four waves: T1: March–April 2002 T2: September–October 2002 T3: April–May 2003 T4: October–November 2003	T1: 142 T2: not reported T3: not reported T4: 79	**Organizational context** -**Innovation**–negative relation with EE and DP, and positive relation with PA–P -**Work pressure (climate)**–positive relation with EE–D **Individual characteristics** -**Neuroticism**–positive relation with EE, and negative relation with PA–D	**External validity** *Major flaws*: -Exclusion rate from the analysis >10% *Minor flaws*: -Non-general population -Self-selection of participants -Incomplete justifications of the sample size **Internal validity** *Minor flaws*: -Some subscales of the work climate scale with low reliability coefficients
González-Morales et al. [53], Anxiety, Stress and Coping, 2012, Spain * EE and cynicism	Two waves: during the first term and again six or seven months later during the third and last term of the academic year	T1: 659 T2: 555	**Organizational context** -**Perceived collective EE and cynicism at T1** positively predicted individual EE and cynicism at T2, respectively–D -**Teacher-student ratio** negatively predicted cynicism at T2–P **Individual characteristics** -**Individual EE and cynicism at T1** positively predicted EE and cynicism at T2, respectively–D -**Individual collective cynicism at T1** positively predicted individual cynicism at T2–D	**External validity** *Minor flaws*: -Non-general population -Self-selection of participants *Poor reporting*: -Not reported exclusion rate **Internal validity** *Minor flaws*: -Reference period different from recommended *Poor reporting*: -Not reported validity and reliability of the Quality of school facilities scale
Houkes et al. [65], Journal of Occupational Health Psychology, 2003, Netherlands EE	Two waves: April 1998 and April 1999	T1: 627 T2: 338	**Organizational context** -T1 **workload** has a negative longitudinal relationship with T2 EE due to negative suppression (generally, positive)–D/P **Individual characteristics** -T1 **negative affectivity** has a negative longitudinal relationship with T2 EE due to negative suppression (generally, positive)–D/P	**External validity** *Major flaws*: -Response rate in total sample <40% -Exclusion rate from the analysis >10% *Minor flaws*: -Non-general population -Self-selection of participants
Innstrand et al. [66], Work and Stress, 2008, Norway EE and disengagement	Two waves: two points in time with a 2 year time interval	T1: 5120 T2: 2235	**Support** -**Work-to-family facilitation** at T1 caused low levels of exhaustion and disengagement at T2–P -High level of **family-to-work facilitation** at T1 predicted low levels of DE at T2–P **Conflict** -**Work-to-family and family-to-work conflict** produced lagged positive effects on EE and DE–D	**External validity** *Major flaws*: -Response rate in total sample <40% *Minor flaws*: -Non-general population -Sampling bias not assessed -Sampling bias not addressed -Self-selection of participants *Poor reporting*: -Exclusion rate from the analysis not reported **Internal validity** *Major flaws*: -Major confounding factors not assessed
Laugaa et al. [68], Revue européenne de psychologie appliqué, 2008, France EE, DP, and Professional non-accomplishment	Two waves: T1 (November–December 2002) and during the second school term (T2: May–June 2003)	T1: 410 T2: 259	**Organizational context** -Positive direct and indirect (perceived stress mediates this) effect of **workload** on EE and RPA–D -Positive indirect effect of **inequity** on EE (perceived stress mediates this)–D -**Perceived stress** has a positive direct effect on EE–D -**Perceived stress** has a negative effect on DP–P -Significant positive indirect effect is observed between **perceived stress** and DP (coping centred on traditional teaching methods mediates this)–D **Individual characteristics** -**Coping centred on the problem**–direct negative effect on all burnout dimensions–P -**Adopting a traditional style of teaching**–direct positive effect on all burnout dimensions–D -**Avoidance coping**–direct positive effect on DP and RPA–D -**Self-efficacy**–direct negative effect on all burnout dimensions–P **Support** -**Social support (satisfaction)**–significant indirect negative effect on the RPA (coping centred on the problem mediates this)–P **Conflict** -**Conflicts and interpersonal problems**–direct positive effect on EE–D	**External validity** *Major flaws*: -Response rate in total sample <40% -Exclusion rate from the analysis >10% *Minor flaws*: -Non-general population -Self-selection of participants **Internal validity** *Minor flaws*: -Reference period different from recommended
Prieto et al. [69], Psicothema, 2008, Spain EE, DP, and cynicism	Two waves: T1 at the beginning of the academic year and eight months later at the end of the academic year (T2)	T1: 484 T2: 274	**Organizational context** -**Quantitative overload** is a good positive predictor of EE at T2–D **Individual characteristics** -Only **gender** and quantitative overload show main effects, irrespectively of the level of EE (women)–D -When controlling by baseline level of cynicism at T1 only **gender** and **role conflict** continue to be significant predictors of cynicism over the time (women)–D **Conflict** -**Role conflict** is a good positive predictor of cynicism at T2–D	**External validity** *Minor flaws*: -Non-general population -Self-selection of participants *Poor reporting*: -Exclusion rate from the analysis not reported **Internal validity** *Minor flaws*: -Reference period different from recommended
Malinen et al. [51], Teaching and Teacher Education, 2016, Finland* EE	Three waves: late September 2013, late January 2014, and late April 2014	T1: 571 T2: 472 T3: 486 365 at all waves	**Organizational context** -**General school climate** had a negative indirect effect on burnout–P **Individual characteristics** -**Job satisfaction**–direct negative effect on burnout–P -**Teacher self-efficacy**–direct negative effect on burnout–P	**External validity** *Minor flaws*: -Non-general population -Self-selection of participants -Sampling bias not assessed, but justified the missing data *Poor reporting*: -Exclusion rate from the analysis not reported **Internal validity** *Major flaws*: -Major confounding factors not assessed *Minor flaws*: -Reference period different from recommended -Low reliability of the Decision making subscale from the School climate scale
Mauno et al. [70], Work and Stress, 2015, Finland EE	Three waves: Time 1: 2008 Time 2: 2009 Time 3: 2010, each time in the autumn	T1: 2137 T2: 1314 T3: 926 Final sample: 814	**Organizational context** -Temporary workers (**type of contract**) reported more EE at each wave. However, the Group × Time interaction effect was not fully consistent–D **Support** -In the absence of **work-family enrichment**, temporary employees reported the highest level of EE. The level of WFE did not affect the level of EE in permanent employees over time–D	**External validity** *Major flaws*: -Exclusion rate from the analysis >10% *Minor flaws*: -Non-general population -Self-selection of participants
Parker et al. [71], Teaching and Teacher Education, 2012, Australia EE, DP and PA combined in a single measure	Two waves: Not reported reference period	T1: 778 T2: 430	**Individual characteristics** -**Mastery** was a strong positive predictor of problem-focused coping at both time waves and a significant negative predictor of emotion-focused coping–P -**Failure avoidance** was a strong predictor of emotion-focused coping and negative predictor of problem-focused coping–D -**Problem-focused coping** was a negative predictor of burnout–P -**Emotion-focused coping** was a strong positive predictor of teacher burnout–D	**External validity** *Minor flaws*: -Non-general population -Self-selection of participants *Poor reporting*: -Exclusion rate from the analysis not reported **Internal validity** *Poor reporting*: -Reference period not reported
Philipp et al. [72], Journal of Occupational Health Psychology, 2010, Germany EE	Two waves: two points in time with a 10 month time interval	T1: 210 T2: 102	**Individual characteristics** -**Deep acting** had a negative effect on EE over the period of a year–P	**External validity** *Major flaws*: -Response rate in total sample <40% -Exclusion rate from the analysis >10% *Minor flaws*: -Non-general population -Self-selection of participants **Internal validity** *Major flaws*: -Major confounding factors not assessed *Minor flaws*: -Reference period different from recommended
Retelsdorf et al. [73], Learning and Instruction, 2010, Israel EE, DP and Lack of PA into a single score	Two waves: in the first half of the school year and an additional survey at the end of the school year	T1: 78 T2: 69	**Individual characteristics** -**Work-avoidance orientation** is positively related to burnout–D -**Work-avoidance goal orientation** emerged as the only significant predictor of burnout–D	**External validity** *Major flaws*: -Response rate in total sample <40% *Minor flaws*: -Non-general population -Self-selection of participants -Incomplete justifications of the sample size *Poor reporting*: -Exclusion rate from the analysis not reported **Internal validity** *Major flaws*: -Major confounding factors not assessed *Minor flaws*: -Reference period different from recommended
Schwarzer et al. [74], Applied Psychology: An International Review, 2008, Germany EE and DP into a single measure	Two waves: teachers, a part of a nationwide school innovation project called “Self-Efficacious Schools” and one year later	T1: 595 T2: 458	**Individual characteristics** -**Teacher self-efficacy** appears to be a protective resource against job stress, whereas job stress translates directly into burnout (EE and DP)–P	**External validity** *Minor flaws*: -Non-general population -Self-selection of participants -Sampling bias not addressed in the analyses -Sampling bias not assessed *Poor reporting*: -Exclusion rate from the analysis not reported **Internal validity** *Minor flaws*: -Major confounding factors partially assessed
Shirom et al. [75], International Journal of Stress Management, 2009, Israel Burnout	Two waves: T2 questionnaire was administered 7 months later of T1	T1: 1048 T2: 762 Final sample: 404	**Organizational context** -For the stressors of **heterogeneous classes**, the T1 value of this stressor significantly positively predicted T2 burnout–D	**External validity** *Major flaws*: -Exclusion rate from the analysis >10% *Minor flaws*: -Non-general population -Self-selection of participants **Internal validity** *Minor flaws*: -Major confounding factors partially assessed -Reference period different from recommended -Low reliability of the Heterogeneous classes and Physical conditions scales
Tang et al. [2], Journal of Organizational Behavior, 2001, China EE, DP and Lack of PA into a single score	Two waves: January and June, 2000	T1: 83 T2: 72 At both T: 61	**Individual characteristics** -**Burnout at T1** significantly related to burnout at T2–D -**Self-efficacy** negatively related to burnout at T1–P -**Proactive attitude** negatively related to burnout at T1–P	**External validity** *Major flaws*: -Exclusion rate from the analysis >10% *Minor flaws*: -Non-general population -Self-selection of participants -Incomplete justifications of the sample size *Poor reporting*: -Addressing sampling bias not reported -No information about sampling bias **Internal validity** *Minor flaws*: -Reference period different from recommended
Taris et al. [76], Journal of Occupational Health Psychology, 2001, Netherlands EE, DP and Lack of PA	Two waves: Winter, 1996 and winter, 1997	T1: 1309 T2: 998 Final sample: 940	**Conflict** -**Stress experienced in relationship** with the students, colleagues, and organization–positive with EE, DP, and negative with PA (only students and organization)–D -**Perceived inequity** in relationships with students, colleagues, and organization–positive with EE, DP, and negative with PA (only students and organization)–D	**External validity** *Minor flaws*: -Non-general population -Self-selection of participants *Poor reporting*: -Response rate not reported
Taris et al. [77], Anxiety, Stress, and Coping, 2004, Netherlands EE, DP regarding students, DP regarding colleagues, and Reduced PA	Two waves: 1 year between waves	T1: 1309 T2: 998 Final sample: 920	**Conflict** -**Perceived inequity** in relationships with students, colleagues, and organization–small magnitude of association and inconsistent results	**External validity** *Minor flaws*: -Non-general population -Self-selection of participants *Poor reporting*: -Response rate not reported **Internal validity** *Major flaws*: -Major confounding factors not assessed
Vera et al. [78], Estudios de Psicología, 2012, Spain EE, cynicism, and DP into a single score	Two waves: T1 at the beginning of the academic year, and the second one (T2) eight months later at the end of the academic year	T1: 484 T2: 274	**Organizational context** -**Job demands** (overload and role conflict positively predicted burnout–D -**Job resources** (autonomy and climate) negatively predicted burnout–P **Individual characteristics** -**Self-efficacy** at T1 was negatively related to burnout at T2–P -Also, self-efficacy showed mediation effect between JDs and burnout–P	**External validity** *Minor flaws*: -Non-general population -Self-selection of participants *Poor reporting*: -Exclusion rate from the analysis not reported **Internal validity** *Major flaws*: -Major confounding factors not assessed *Minor flaws*: -Reference period different from recommended
Salanova et al. [49], Revista de Psychologia, 2005, Spain* EE, DP, and cynicism	Two waves: T1 (in the beginning of academic year) and T2 (in the end of the academic year)	T1: 438 T2: 274	**Organizational context** -**Effective class management**–negative and significantly associated with EE and cynicism but these associations are mediated by EE and cynicism in T1–P **Individual characteristics** -**Burnout dimensions at T1**–mediating in all cases the relationship among obstacles/facilitators and burnout in T2–D -**Gender**–woman more exhausted than men–D **Conflict** -**Social obstacles regarding to students and parents**–positive and significantly associated to cynicism and DP but these associations are mediated by DP in T1–D	**External validity** *Minor flaws*: -Non-general population -Self-selection of participants -Sampling bias not addressed in the analyses -Sampling bias not assessed *Poor reporting*: -Response rate in total sample not reported -Exclusion rate from the analysis not reported -Number of screened and eligible not reported **Internal validity** *Minor flaws*: -Reference period different from recommended *Poor reporting*: -Validity of independent variables not reported
Laugaa et al. [67], L’orientation scolaire et professionnelle, 2005, France EE, DP, and professional nonaccomplishment	Two waves: T1: November 2002 (first trimester) T2: third trimester	T1: 410 T2: 259	**Individual characteristics** -**Problem-focused coping** proved to be functional by attenuating burnout at T2 (reduces EE and DP)–P -**Avoidance coping** had a deleterious effect by worsening subsequent burnout (increases DP and RPA)–D -**Conservative (traditional) pedagogical style** had a deleterious effect by worsening subsequent burnout (increases DP and RPA)–D	**External validity** *Minor flaws*: -Non-general population -Self-selection of participants -Sampling bias not addressed in the analyses -Sampling bias not assessed *Poor reporting*: -Response rate in total sample not reported -Exclusion rate from the analysis not reported -Not reported sampling method -Number of screened and eligible not reported **Internal validity** *Minor flaws*: -Major confounding factors partially assessed -Reference period different from recommended -Did not obtain methods to reduce bias
Llorens et al. [79], Revista de Psicología del Trabajo y de las Organizaciones, 2005, Spain EE, DP, and cynicism into a single score	Two waves: T1 (in the beginning of academic year) and T2 (in the end of the academic year)	T1: 484 T2: 274	**Individual characteristics** -**Reduced self-efficacy** increases burnout–D **Conflict** -**Exposure to obstacles** enhance a lack of perceived efficacy that in turn enhances burnout (EE and cynicism)–D	**External validity** *Minor flaws*: -Non-general population -Self-selection of participants *Poor reporting*: -Response rate in total sample not reported -Exclusion rate from the analysis not reported -Number of screened and eligible not reported **Internal validity** *Minor flaws*: -Reference period different from recommended *Poor reporting*: -Validity of independent variables not reported

* Papers on which we conducted quantitative analysis. D—Detrimental effects; P—protective effects; EE—emotional exhaustion; DP—depersonalization; PA–personal accomplishment.

**Table 2 ijerph-19-05776-t002:** List of burnout determinants detected within the qualitative analysis.

Determinants	Explanation	Effects (Detrimental D or Protective P), Relationship with Job Demands (JD) or Job Resources (JR) of JD/JR Model of Burnout *	Study (1st Author, Year of Publication)
Support			
Lack of social integration	-Marital status and having children [48]	D, lack of JR	Burke et al., 1996 [54]
Lack of social support/Lack of work-family enrichment	-Lack of maintaining friendships outside of work [47]; -Lack of the positive effects of work/family quality on the family/work quality [60]	D, lack of JR	Burke et al., 1995 [58] Mauno et al., 2015 [70]
Social support from colleagues	-Inside the school: refers to all other workers in the school [37]; -Outside the school: refers to colleagues from other schools [37]; -The extent to which each of the items (e.g., harmonious ,enriching, satisfying, and inspired trust) corresponded to the current relationships with co-workers [51]	P, JR	Beausaert et al., 2016 [47] Fernet et al., 2010 [61]
Social support from the broader community	-Refers to the broader professional network, not only including other principals, but also teachers, counsellors, parents, and community leader [37]; -Emotional support from the broader community [38]; -Satisfaction in relation to the available social support [58]	D or P, JR	Beausaert et al., 2016 [47] Feuerhahn et al., 2013 [48] Laugaa et al., 2008 [68]
Work-to-family/Family-to-work facilitation	-The extent to which the skills, behaviours, positive mood, and support or resources from one role positively influenced the other role [56]	P, JR	Innstrand et al., 2008 [66]
**Conflict**			
Parental criticism	-Parents of the pupils criticizing the teachers’ work [38]; -Social barriers due to relationship with parents [39,70]	D, JD (Emotional)	Feuerhahn et al., 2013 [48] Salanova et al., 2005 [49] Llorens et al., 2005 [79]
Conflicts and interpersonal problems	-Refers to conflicts or strained relations with parents; lack of respect, arrogance or violence on the part of some students; being blamed by some parents for their child’s scholastic difficulties; pressure from the school inspector to improve their work or to work differently; fear of committing a professional error [58]	D, JD (Emotional)	Laugaa et al., 2008 [68]
Disruptive students/ Classroom interruptions	-Refers to the difficulties in controlling the class, meeting uncooperative and troublemaking students, and impatience when students do not do what they are asked to do [48]; -Students do not pay attention to the content of lessons and disturb lessons [38]	D, JD (Emotional)	Burke et al., 1996 [54] Feuerhahn et al., 2013 [48]
Perceived inequity in relationships with colleagues	-Refers to the comparison between investments in the work relationship with colleagues and benefits from this relation [67]	D, JD (Emotional)	Taris et al., 2001 [76]
Perceived inequity in relationships with organization	-Refers to the comparison between investments in the work relationship with school management and benefits from this relation [67]	D, JD (Emotional)	Taris et al., 2001 [76]
Perceived inequity in relationships with students	-Refers to the comparison between investments in the work relationship with students and benefits from this relation [67]	D, JD (Emotional)	Taris et al., 2001 [76]
Stress due to societal demands	-Expectations from society on professors and the educational system [49]	D, JD (Emotional)	Buunk et al., 2007 [59]
Stress due to relationship with students	-Relationships with students including the diversity of the students [49]; -Lack of interest and motivation in students and misbehaviour among students [67]; -Disinterest and lack of motivation among students for learning, and low discipline [39,70]	D, JD (Emotional)	Buunk et al., 2007 [59] Taris et al., 2001 [76] Salanova et al., 2005 [49] Llorens et al., 2005 [79]
Stress due to relationship with colleagues	-Refers to incompetent colleagues and colleagues who do not adhere to mutual agreements [67]	D, JD (Emotional)	Taris et al., 2001 [76]
Stress due to relationship with organization	-Refers to not functional school management [67]	D, JD (Emotional)	Taris et al., 2001 [76]
Work-to-family/Family-to-work conflict	-The extent to which time pressures and strain in one role interfered with performance in the other role [56]	D, JD (Emotional)	Innstrand et al., 2008 [66]
**Individual characteristics**			
Adopting a traditional style of teaching	-Maintaining discipline, punishing students, insisting that the students remain quiet, behaving in an authoritarian manner, separating or isolating certain students from the others for a while, keeping the students busy, developing habits in the way of teaching [57,58]	D, Neither	Laugaa et al., 2005 [67] Laugaa et al., 2008 [68]
Burnout symptoms/Individual burnout at T1	-Emotional exhaustion and depersonalization subscales at T1 [39,44,66]; -Exhaustion and cynicism at T1 [43]	D, Neither	Bianchi et al., 2015 [55] González-Morales et al., 2012 [53] Tang et al., 2001 [2] Salanova et al., 2005 [49]
Coping–Avoidance coping/Work-avoidance goal orientation	-Not bringing work home, completely forgetting work when the day is over, neither working too hard nor too long, getting more involved in extraprofessional activities, simply attempting to ignore the problems, avoiding the other members of the teaching staff, telling yourself that it is just a job and continuing to do it [57,58]; -Tendency to focus on attempting to avoid, deflect, or re-interpret the implications of demands for self-esteem and self-worth [61]; -Refers to strivings to get through the day with little effort [63]	D, Neither	Laugaa et al., 2005 [67] Parker et al., 2012 [71] Retelsdorf et al., 2010 [73] Laugaa et al., 2008 [68]
Coping–Centred on the problem	-Attempting to objectively analyze the situation and controlling one’s emotions, thinking about the positive aspects of teaching, taking stock of the situation and attempting to rationalize it, giving the students positive encouragement, attempting to always remain coherent and honest in the relation with the students [57,58]; -Refers to persistence (keep trying at difficult things in work), planning (usually stick to a work timetable or work plan), and self-management (usually tries to find a place where one can prepare well) [61]	P, Neither	Laugaa et al., 2005 [67] Parker et al., 2012 [71] Laugaa et al., 2008 [68]
Coping–Direct coping style	-A problem-solving behaviour through rational and task-oriented strategies [50]	P, Neither	Carmona et al., 2006 [60]
Coping–Emotion-focused coping	-Refers to strategies directed toward reinterpreting or changing the meaning of threats and challenges [61]	D, Neither	Parker et al., 2012 [71]
Deep acting	-Regulating feelings by individuals and actually changing their inner emotional state in order to really feel the appropriate emotion [62]	P, Neither	Philipp et al., 2010 [72]
Downward identification	-An individual views oneself as similar to others who are functioning in a worse way, or that one views the situation of worse-off others as a possible future for oneself, which will generally induce negative feelings [50]	D, Neither	Carmona et al., 2006 [60]
Individual stress	-Refers to how tense, irritable and stressed people where [37]	D, Neither	Beausaert et al., 2016 [47]
Individual demographic characteristics (including gender)	-Age, work experience, marital status, gender, position, level of education, children [46]; -Gender [59]; -Age, gender [39]	D, Neither	Burke et al., 1995 [57] Prieto et al., 2008 [59] Salanova et al., 2005 [49]
Interpersonal rejection sensitivity	-Particular sensitivity to another person’s judgment and criticism, with the recurrent fear of being rejected (this resulting ,for instance, in stormy relationships, inability to sustain long-term relationships, problems at work, difficulties initiating contacts ,pervasive fear of embarrassment) [44]	D, Neither	Bianchi et al., 2015 [55]
Job satisfaction	-Satisfaction with the current job (happiness to come to work, and to continue for a long time in the current workplace; the rewarding nature of current job; and enjoying of being in the current job position [41]	P, Neither	Malinen et al., 2016 [51]
Loss of status	-The frequency of encountering a series of experiences, including: things that damaged one’s reputation, or feelings of lost status, power or influence [49]	D, Neither	Buunk et al., 2007 [59]
Mastery	-Seeing obstacles as malleable and, as such is associated with the perception that demands are responsive to task-directed effort and/or strategy [61]	P, Neither	Parker et al., 2012 [71]
Negative affectivity	-Refers to the extent to which participants, in general, experienced several mood states (guilty, ashamed, nervous, and distressed) [55]	D or P, Neither	Houkes et al., 2003 [65]
Neuroticism	-Neuroticism as the disposition to interpret events negatively [42]	D, Neither	Goddard et al., 2006 [52]
Passion–Harmonious passion	-Passion for teaching characterized by strong psychological investment in a passionate activity that has been autonomously internalized within the identity; the activity is under the control of the individual [52]	P, Neither	Fernet et al., 2014 [62]
Passion–Obsessive passion	-Passion that results from controlled internalization of an activity within the individual’s identity. The investment in the activity gets out of the individuals’ control; the activity controls the individual [52]	D, Neither	Fernet et al., 2014 [62]
Perceived self-efficacy	-Perceived self-efficacy in classroom management and techniques in managing student behavior [38,41,45,58,69]; -Refers to job accomplishment, skill development on the job, social interaction with students, parents, and colleagues, and coping with job stress [64]; -Assessing the strength of people’s belief in their own abilities to respond to novel or difficult situations and to deal with any associated obstacles [66]	P, Neither	Browers et al., 2000 [56] Feuerhahn et al., 2013 [48] Laugaa et al., 2008 [68] Malinen et al., 2016 [51] Schwarzer et al., 2008 [74] Tang et al., 2001 [2] Vera et al., 2012 [78]
Proactive attitude	-Measuring people’s belief in the rich potential of changes that can be made to improve themselves and their environment [66]	P, Neither	Tang et al., 2001 [2]
Reduced self-efficacy	-Refers to the dimension of professional efficacy of the MBI-GS questionnaire at T1 [70]	D, Neither	Llorens et al., 2005 [79]
Self-determined work motivation	-Includes subscales on intrinsic motivation, identified regulation, introjected regulation, and external regulation combined into a composite score [51]	P, Neither	Fernet et al., 2010 [61]
Self-doubts	-Self-doubts in coping with professional demands [48]	D, Neither	Burke et al., 1996 [54]
Self-critical form of perfectionism	-Refers to feelings of uncertainty regarding the quality of everyday actions and a vague sense that tasks have not been satisfactorily completed [54]	D, Neither	Flaxman et al., 2012 [64]
Sense of defeat	-Refers to feelings such as: not made it in life, completely knocked out of action, or having lost important battles in life [49]	D, Neither	Buunk et al., 2007 [59]
Unmet expectations	-Refers to the amount of fulfilment of expectations [46]	D, Neither	Burke et al., 1995 [57]
Work-related worry and rumination	-Refers to the features of perseverative cognition: cognitive content that focused explicitly on (work-related) stressors or problems; a degree of repetitive and uncontrollable thinking; and a focus on potentially negative outcomes occurring in the past and/or future [54]	D, Neither	Flaxman et al., 2012 [64]
**Organizational context**			
Ambiguity/conflict	-Unclear roles and ambiguity in job tasks [46,59,69]	D, JD (Organizational)	Burke et al., 1995 [57] Prieto et al., 2008 [69] Vera et al., 2012 [78]
Autonomy	-Autonomy in decision making [69]	P, JR	Vera et al., 2012 [78]
Effective class management	-Refers to possibility to change the type or dynamics of class activities, access to information and materials for class, or use the humor in class [39]	P, JR	Salanova et al., 2005 [49]
Heterogeneous classes	-Refers to heterogeneous classes in which it was difficult to adapt the level of instruction to students’ instructional needs, and in large classes in which it was difficult to provide individual attention to students [65]	D, JD (Physical or Organizational)	Shirom et al., 2009 [75]
Inequity	-Refers to lack of consideration for the job of teaching, little perspective of career advancement and promotions, an inadequate salary in light of the responsibilities and the work put in, the lack of recognition for the work and the efforts put in, the inflexibility of working hours [58]	D, JD (Emotional)	Laugaa et al., 2008 [68]
Innovation	-The perception of work climate as rich with workplace innovation; how innovative the school environment was [42]	P, JR	Goddard et al., 2006 [52]
Lack of stimulation	-Refers to challenging and stimulating nature of the job [46]	D, JD (Cognitive)	Burke et al., 1995 [57]
Lack of social support (via sources of stress)	-Refers to support out of work [47]	D, JD (Emotional) or lack of JR	Burke et al., 1995 [58]
Narrow client contacts	-Frequent direct contact with other people [46]	D, JD (Emotional or Physical–demanding contacts)	Burke et al., 1995 [57]
Perceived collective burnout at T1	-Collective burnout that reflects the perceptions of the individual about his or her colleagues’ burnout symptoms [43]	D, Neither	González-Morales et al., 2012 [53]
Red tape work	-Refers to bureaucratic work and conflicts with rules and procedures, unnecessary regulations, and conflicts between school rules and students’ needs [48]	D, JD (Organizational)	Burke et al., 1996 [54]
School climate	-School climate related to collaboration, student relations, decision making, and instructional innovation [41]; -Refers to support climate (helping each other), goals climate (clear targets to be achieved over a period of time), innovation climate (new ideas implemented), rules climate (highly regulated work) [69]	P, JR	Malinen et al., 2016 [51] Vera et al., 2012 [78]
Sources of stress	-Refers to doubts about competence, problems with clients, bureaucratic interference, and lack of fulfilment and collegiality [47] -Stress due to role problems, career and achievement, professional relationships and relationships with students [49] -Conflicts and interpersonal problems ,workload, professional non-accomplishment, and inequity [58]	D, JD (All types)	Burke et al., 1995 [58] Buunk et al., 2007 [59] Laugaa et al., 2008 [68]
Teacher-student ratio	-Indicator of demands computed by dividing the number of teachers in a school by the number of students in the school [43]	P, JD (low Physical JD)	González-Morales et al., 2012 [53]
Time pressure/Work pressure (climate)/Workload	-Refers to frequency of being pressed for time at work [38]; -The perception of work climate where workers experience significantly high work pressures [42]; -Quantitative and qualitative demanding aspects in the work situation, such as working under time pressure, working hard, and strenuous work [55]; -Refers to difficulty making progress with children who are failing academically, lack of time to monitor the progress of students individually, feeling responsible for their students’ results, having too many things to do and not enough time to do everything, heavy workload [58]; -Refers to quantitative overload and too much work to do [59,69]	D, JD (Physical)	Feuerhahn et al., 2013 [48] Goddard et al., 2006 [52] Houkes et al., 2003 [65] Laugaa et al., 2008 [68] Prieto et al., 2008 [69] Vera et al., 2012 [78]
Type of contract (temporary workers)	-Refers to temporary and permanent workers [60]	D, JD (Organizational)	Mauno et al., 2015 [70]
Type of school	-Elementary, junior high or secondary [46]	D, JD (Organizational or Physical)	Burke et al., 1995 [57]
Work setting characteristics	-Refers to inadequate orientation, workload, lack of stimulation and autonomy, unclear goals, poor leadership, and social isolation [47]	D, JD (Physical)	Burke et al., 1995 [58]

* D—Detrimental effects; P—protective effects; JD—job demands; JR–job resources.

**Table 3 ijerph-19-05776-t003:** List of burnout determinants analyzed via quantitative synthesis.

Determinants	Articles Including the Determinant
Support	
From colleagues	Beausaert et al., 2016 [47]
From supervisor	Beausaert et al., 2016 [47]
From community	Beausaert et al., 2016 [47]
Emotional support	Feuerhahn et al., 2013 [48]
Social facilitators	Salanova et al., 2005 [49]
**Conflict**	
With colleagues	Feuerhahn et al., 2013 [48]
Emotional strain	Feuerhahn et al., 2013 [48]
Parent criticism	Feuerhahn et al., 2013 [48]
Obstacles parents/students	Salanova et al., 2005 [49]
**Individual characteristics**	
Emotional dissonance	Feuerhahn et al., 2013 [48]
Teacher self-efficacy	Feuerhahn et al., 2013 [48] Malinen et al., 2016 [51]
Exhaustion, depersonalization, and/or cynicism at T1	Salanova et al., 2005 [49]
Neuroticism	Goddard et al., 2006 [52]
Job satisfaction	Malinen et al., 2016 [51]
**Organizational context**	
Time pressure	Feuerhahn et al., 2013 [48]
Classroom disruption	Feuerhahn et al., 2013 [48]
Perceived collective exhaustion	Gonzalez-Morales et al., 2012 [53]
Perceived collective cynicism	Gonzalez-Morales et al., 2012 [53]
Workload stressors	Gonzalez-Morales et al., 2012 [53]
Technical obstacles	Salanova et al., 2005 [49]
Effective class management	Salanova et al., 2005 [49]
Work climate	Goddard et al., 2006 [52]
School climate	Malinen et al., 2016 [51]
Collective teacher efficacy	Malinen et al., 2016 [51]

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
