# Peer review of "Determinants of Burnout among Teachers: A Systematic Review of Longitudinal Studies"

_ijerph, 2022, doi:10.3390/ijerph19095776_

Round 1
Reviewer 1 Report
I find the paper quite well written, addressing properly the topic. My only remarks are:
- it is not clear why the authors conducted the study only until 2018 and not until 2021 or 2022.
- some more references on the topic could be added.
Author Response
Respected Reviewer,
Thank you for the comments and suggestions.
- We have added a sentence within the Methods section in order to clarify that the literature search was extended up to December 2021: "Because there was possibility that additional studies were published during or after finalizing this review, we checked databases for new publications up to December 2021 and no additional prospective longitudinal studies on burnout in teachers were identified."
- About the citation of two additional studies, we have decided to not include them in this manuscript because they are related to gender differences in certain psychological phenomena during the pandemic which is not a focus of the current systematic review.
Regards,
Prof. Mijakoski
Reviewer 2 Report
The authors presented a systematic review of longitudinal studies aimed at identifying determinants of teachers’ burnout. The literature analysis is rigorous with respect to research protocols and well described in the paper. The results section is very rich of information, even though the data presentation in the tables make the article’s reading a little heavy. The work is very interesting and worthy of publication, nevertheless some improvements could be implemented to improve its quality:
- accurate mother tongue revision;
- streamline the information in Tables 1 and Table 2, using more concise wording (e.g. Table2 “refers to not functional school management” -> “not functional school management” etc.…); table 3 should have the same layout of the previous tables. Probably, a horizontal layout for the tables may enhance the readability of the text.
Author Response
Respected Reviewer,
Thank you for the comments and suggestions:
- The paper has been reviewed by a co-author who is a native English speaker;
- We have streamlined the information by adding some explanations, such as: "Table 1 refers to the characteristics of 33 studies on burnout among teachers that were reviewed."; "Table 2 refers to the list of burnout determinants that were detected within qualitative analysis."; and "Table 3 refers to the list of burnout determinants that were analyzed with quantitative synthesis."
- We have produced a new layout of Table 3.
Regards,
Prof. Mijakoski
Reviewer 3 Report
This is a review of the manuscript: Determinants of burnout among teachers: a systematic review of longitudinal studies. The authors have conducted a systematic literature review on longitudinal studies measuring teachers’ burnout as a dependent variable. A qualitative and a quantitative review were presented.
Critical view of the results is a key strong point of the manuscript. Several areas require minor improvements.
- Introduction needs to highlight mote the need for reviewing longitudinal studies on burnout. This is a good asset of the work, so it needs to be emphasized.
- In subsection 1.1., paragraph 2: other categories of teachers could be described too since they are referred to in the categorizations of ILO.
- The work would be benefited if for each study you present the level of education too (primary, secondary etc). This would also help in interpreting some of the findings, as different settings have different occupational cultures.
- In subsection 1.1., paragraph 3: you can talk here about psychosocial risks too.
- Please refer to JD-R model by name.
- Objective should be updated to include your focus on longitudinal research. In the objective your argument in terms of qualitative and quantitate analysis is not clear.
- You need to justify your choice of including studies with sample size larger than 50 (I guess power, but needs to be included). Also please clarify whether 50 is the final wave sample.
- Subsection 2.4: it would be better to structure this section per determinant, so that the reader has first all the information for support, then for conflict etc.
- PRISMA should be in the method section.
- In table 2 you can list the determinants in alphabetical order (clustered in the 4 categories as you already do) or use another meaningful order, so that the reader can either easily locate them or get a better understanding of them.
- It would be good to clearly report which statistical method the longitudinal studies used.
- In the first paragraph of the discussion as well as your conclusion please summarize the results of your qualitative analysis too.
- Please include practical implications of your work.
Author Response
Respected Reviewer,
Thank you for the valuable comments and suggestions. Please find the corrections in the text (marked with yellow highlight):
- We have highlighted the need for reviewing longitudinal studies on burnout (Introduction section);
- In subsection 1.1., paragraph 2, we have briefly described other categories of teachers;
- In subsection 1.1., paragraph 3, we have added a part about psycho-social hazards;
- We have referred to JD-R model by name;
- The Objective has been updated with included focus on longitudinal research;
- In order to justify the choice of including studies with sample size larger than 50, we have put: "(i.e., studies with sufficient power)". We have also clarified that 50 was the final wave sample;
- Subsection 2.4 is structured per determinant;
- We have put the PRISMA flow-chart within the Results section since subsection 3.1 gives the description of selected studies as a "result" of the review;
- We have arranged the determinants in table 2 in alphabetical order (clustered in 4 categories);
- We have not reported the statistical method the longitudinal studies used because of the robustness of the Table 1;
- In the first paragraph of the Discussion as well as in the Conclusion, we have put only the determinants of exhaustion identified by quantitative synthesis (according to their relative importance - effect size), because those factors showed the strongest evidence of predicting exhaustion. We have also taken into consideration the findings of the risk of bias assessment of the studies included. Therefore, the findings of the qualitative analysis were summarized in the following section of the Discussion (4.1. Interpretation of findings);
- The practical implications of the work have been included in the Discussion section.
Regards,
Prof. Mijakoski